# Challenges and opportunities for integrating traditional healing approaches with biomedical care for mental illness: A scoping review from healers' perspectives

**Alemayehu Molla Wollie**[1,2]*, **Kim Usher**[1], **Kylie Rice**[3], **Md Shahidul Islam**[1]

**1** School of Health, Faculty of Medicine and Health, University of New England, Armidale, New South Wales, Australia, **2** Department of Psychiatry, College of Medicine and Health Sciences, Injibara University, Injibara, Ethiopia, **3** School of Psychology, Faculty of Medicine and Health, University of New England, Armidale, New South Wales, Australia

* alexmolla09@gmail.com

## Abstract

### Background

A considerable number of people receive care from traditional healers for mental illness. Integrating traditional healing practices with modern treatment may improve the outcomes for people with the symptoms of mental illness. However, there has been limited joint efforts toward integration of the two approaches. Therefore, this review is intended to summarize the challenges and opportunities of integrating traditional treatment approaches with biomedical treatment for mental illness from the perspective of traditional healers.

### Methods

Asksey and O'Malley's framework and the Preferred Reporting Items for Systematic Review and Meta-Analysis Extension for Scoping Review (PRISM-ScR) guidelines were followed to conduct this scoping review. Searches of databases, including PubMed/Medline, PsycINFO, CINAHL, Scopus, and the Web of Sciences were conducted. Additionally, Google and Google Scholar were searched for other information, including grey literature. All articles published between January 2014 and June 2024 were considered. Themes and subthemes were created using Nvivo-12 software. A thematic synthesis was used to report the evidence.

### Result

Based on the eligibility criteria, 54 of the 4071 initially identified articles were included. From the healers' perspectives, governmental policy (guidelines, recognition, education, training, and financial issues), intellectual property issues, attitude,

**Data availability statement:** All relevant data are within the article.

**Funding:** The author(s) received no specific funding for this work.

**Competing interests:** The authors have declared that no competing interests exist.

disease understanding differences, and referral are the major challenges for the integration of traditional healing approaches with biomedical treatment for mental illness. Healers' willingness for integration, the effectiveness of the healing approaches, and the World Health Organization's recognition of traditional healing are identified as enablers for integration.

## Conclusion

Integrative work between traditional healing approaches and biomedical services presents an opportunity to assist in addressing the treatment gap for mental illness. This review presents a synthesis of the major challenges that hinder the integration of traditional healing approaches with biomedical care, and enablers that may facilitate integration. This review of the evidence can support policymakers and other stakeholders in reducing the major challenges of integration noted by healers and maximising opportunities for collaboration. The review also highlights the need to design culturally appropriate guidelines for integration and referral between the two systems.

## Background

Traditional healing practice is a body of knowledge and a set of beliefs that use culturally accepted spiritual treatments and plant products to identify, treat, and prevent disease and preserve health [1,2]. Traditional treatments include a broad range of practices commonly embedded within contextual cultural milieus, reflecting community beliefs [2,3]. The preference for traditional healing methods is increasing in many countries [2,4], with a significant number of people who have mental illness choosing traditional treatments [5,6], mainly from low-and middle-income countries [7,8]. High-income countries also use it as a complementary or alternative approach, often alongside available modern treatments [9]. For many people, traditional treatment is the first option, and practitioners have played an essential role in treating chronic illnesses, including mental illness [10–13].

In low-income countries, due to a lack of mental health specialists and unaffordable biomedical care, nearly half of the population seeks traditional healers for their mental illnesses [14,15]. In addition, superstition related to mental illness is another leading factor in people's preference for traditional healing approaches [16–18]. For example, most people in low-income countries view good health as a gift from God [19], and believe God provides healing in many ways [20,21]. Traditional healers are known to have cultural continuity and use practices based on naturally acquired knowledge [22,23]. Traditional healing practices may include massage, faith or spiritual therapies, counselling, ceremonies, and plant-based remedies [24–27]. Some traditional healers are well-known religious people who employ culturally specific techniques for the symptoms of mental illness [15,28]. Evidence indicates that traditional healers may provide effective psychosocial intervention, particularly for common mental disorders such as depression and anxiety [29,30].

### Rationale

Effective integrative care requires good communication between traditional healers and mental health professionals, which is promising in filling the gap, particularly in resource-limited countries [2,31,32]. Integrating traditional healing practices with modern treatment has also been shown to improve the outcomes of people with symptoms of mental illness [2,33]. In contemporary literature, there is also some awareness of the value in integrating traditional healing approaches with modern bio-psychosocial approaches for mental health care [33–35]. Despite this recognition that integration is important, there is a limitation on collaborative work between the two treatment approaches due to different challenges. Scientific validation of issues of traditional medicine, adapting cultural knowledge to modern life, and lack of standardization were some of the challenges for integration among high-income countries [36,37]. In low-income countries, conceptual understanding differences, attitudes, stigma, limitations in infrastructure, and related issues were commonly reported as challenges of integration [38–42]. For example, attitudinal differences or mistrust of traditional healing approaches by modern health professionals were mentioned in studies [43,44]. In addition, there is also conflicting evidence regarding the positions of traditional healers toward integration. At the same time, some are willing to consult health professionals if they have a problem that they can't solve, but significant numbers of healers have a gap in sharing their experiences with health professionals [45]. Sometimes, traditional healers believe that their treatment approaches are superior to modern treatment in resolving mental illness [46].

The World Health Organization's (WHO) acknowledgment of using evidence-based traditional medicine [47] and the large involvement of service users in traditional healing approaches urges exploring systematic challenges and opportunities for integrative treatment approaches. Additionally, there is no compiled evidence on the challenges and opportunities related to integrating traditional healing practices with modern mental health care from the healers' perspectives. Furthermore, challenges for integration were not consistent within independent studies. Therefore, the objective of this scoping review was to summarize challenges and opportunities of integrating traditional treatment approaches with biomedical treatment for mental illness from the perspective of traditional healers.

## Methods and materials

The review followed Arksey and O'Malley's framework (2005) (identifying research questions, identifying essential studies, selecting studies, charting data, and summarizing and reporting results). The Preferred Reporting Items for Systematic Review and Meta-Analysis Extension for Scoping Review (PRISMA.ScR) Guidelines were followed [48]. The framework of population (traditional healers), concept (challenges and/or opportunities to integration), and context (global) were used to design the eligibility criteria of articles for this review.

### Inclusion criteria

Traditional healers (faith or religious healers, herbalists, for example), as defined by [29], were considered for this scoping review. This scoping review is aimed at addressing summarized evidence in a global context on the challenges and/or opportunities of integrating traditional healing with biomedical treatment for mental illness. All published and unpublished articles were included; as a result, the quality of the articles are not the focus of this review. All studies published in English and released between January 2014 and June 30, 2024, were included in this review to incorporate contemporary evidence.

### Exclusion criteria

Studies that did not incorporate traditional healers and did not include mental illness were excluded from this review. In addition, articles that did not focus on challenges and/or opportunities and integration were not considered. Non-English publications, protocols, commentaries, editorials, news items, letters, and publications before 2014 were excluded from the study.

## Searching strategy

A search strategy was developed by consulting senior librarians from the University of New England Dixson Library. Comprehensive lists of primary keywords and additional synonyms words were generated from the research questions. The final search strategy was applied connecting keywords by Boolean operators on databases like PubMed, Scopus, PsycINFO, Web of Sciences, and CINAHL. Primary searching keywords were challenges, opportunities, traditional healing, integration, modern treatments, and mental illness. Google and Google Scholar were searched manually to ascertain grey literature. Snowball searching of the included articles was conducted to filter the remaining articles.

## Article selection, extraction, and synthesis

The selection process was started by importing all records into an endnote library. Authors screened all relevant articles and confirmed by discussion. At first, article titles and abstracts were screened, keeping those that were relevant to the topic. The full texts of possible papers were then evaluated for eligibility by the same reviewer. After automatically removing duplicates using EndNote and filtering important articles, the data were extracted using the prepared data extraction form. Then, the data results were imported into NVivo-12 software to facilitate coding, creating themes and subthemes, and visualizing created themes. First, frequent readings of data were conducted to become familiar with basic concepts. Then, initial coding and theme generation was conducted by the first author, and other authors confirmed this through continuous meetings. Thematic synthesis was used to summarize the results [49].

# Results

Of the total of 4071 initial outputs, 3886 were identified from databases, while 185 articles were obtained from manual searches on Google and reference lists. After removing duplicates, (n = 3601) were screened in their titles and abstracts, and then 323 full-text articles were evaluated based on eligibility criteria. Finally, 54 articles were included in the final result (Fig 1).

## Overview of included studies

The majority of the included articles were conducted in African countries, and the following studies were conducted in each country: Ghana (n = 11), South Africa (n = 6), India (n = 5), Kenya (n = 4), Uganda (n = 3), Malawi (n = 3), America (n = 3), Liberia (n = 2), Pakistan (n = 2), Tanzania (n = 2), Singapore (n = 2), DR Congo (n = 1), Nigeria (n = 1), Ethiopia (n = 1), Kuwait (n = 1), Nepal (n = 1), United Arab Emirates (n = 1), Zimbabwe (n = 1), Dominican Republic (= 1), Ghana, Nigeria, and Kenya (n = 1). Among the included studies, three were review articles conducted in Ghana, West Africa, and seven low-and middle-income countries [33,50,51] (Table 1).

## The result synthesis of the included studies

**Challenges of integrating traditional healing with biomedical treatment for mental illness from the healers' perspectives.** In accordance with the presentation of the extracted data, the challenges of integration were summarised by five major themes and twelve subthemes. The major themes were attitudes, governmental policy issues, intellectual property, conceptual understanding differences, and referral issues (Fig 2).

**Challenges related to attitudinal issues:** Attitude-related challenges for integration include health professionals' negative attitudes, healers' attitudes, service users' attitudes, and stigma.

**Attitude challenges related to health professionals:** The results of the included 17 studies indicated traditional healers' hold a belief that biomedical practitioners have negative attitudes toward their approaches. The study conducted in Uganda showed that traditional healers believe clinicians have a negative view of them and believe that their works are considered satanic, dirty, and unsanitary [55]. Similarly, the study conducted in South Africa showed that

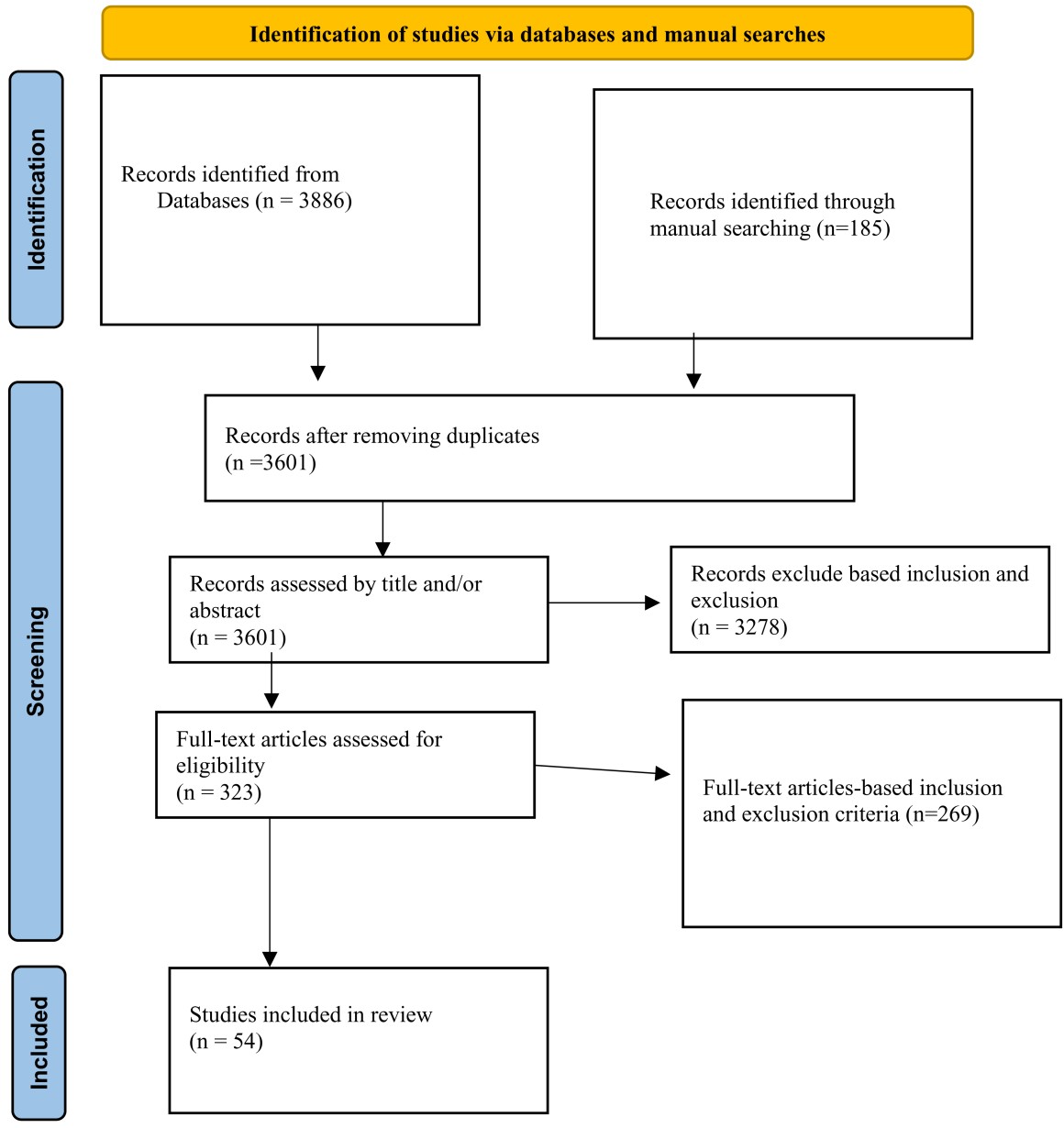

**Fig 1. PRISMA flow diagram; this diagram shows process we followed to select included studies.**

traditional healers felt that health professionals did not respect healers' knowledge [66]. Traditional healers believe that most biomedical practitioners do not want to affiliate with them due to their negative attitudes, according to a study conducted in Liberia [84]. They also believe that health professionals have negative attitudes toward patients whom they consult; this is explained as "the patient was covered with holy mud (Emenet) and the patient looked muddy; on arrival, the doctors chased the patient away, accusing them for being dirty" [34]. Negative attitudes and criticism from medical personnel who do not believe in divine power were mentioned by traditional healers, according to a study conducted in Ghana [61]. Similarly, healers view health practitioners as disrespectful towards traditional healing methods [39–41]. According to study conducted in India, traditional healers feel disrespected and devalued by biomedical doctors, and

**Table 1.  Characteristics of included articles in the scoping review.**

| Author/Year | Country | Aims/objectives | Study Types |
|---|---|---|---|
| Abbo et al., 2019 [52] | Uganda | To examine the relationship between modern medicine and African traditional healing methods. | Qualitative/ natation |
| Ahlberg, 2017 [53] | Kenya | To assess the situation of traditional medicine in central Kenya, evidence from three case studies. | Case study |
| Akol, 2018 [54] | Uganda | To investigate access to mental health services for children and adolescents through primary health care workers and traditional healers. | Mixed method |
| Akol et al., 2018 [55] | Uganda | To explore healers' opinions on their cooperation with modern health systems. | Qualitative |
| Ali, 2023 [56] | India | To explore the interplay between mental health discourse and traditional healing methods. | View Point |
| Almutairi, 2022 [57] | Kuwait | To explore faith healers' views and psychiatrists on cooperating to provide care. | Qualitative |
| Arias et al., 2016 [58] | Ghana | To explore the views and practices of prayer camp staff and biomedical practitioners' perspectives. | Qualitative |
| Arooj, 2023 [59] | Pakistan | Understanding the role of traditional healing practices through different examples. | Narrative |
| Asafo, 2021 [60] | Ghana | To explore social constructions of mental illness within the faith-based healing system. | Qualitative |
| Asamoah et al., 2014 [61] | Ghana | To examine the opinions on the role of churches in mental healthcare delivery. | Qualitative |
| Badu et al., 2019 [50] | Ghana | To identify the pathways used to treat mental illnesses, and evidence about the possibility of collaboration between biomedical, faith, and traditional healing. | Review |
| Baheretibeb et al., 2024 [34] | Ethiopia | To explore holy water priest healers' explanatory models and treatment approaches toward mental illness. | qualitative |
| Bitta et al., 2019 [62] | Kenya | To explore the local terms, perceived causes, and management modalities of the priority mental neurologic and substance disorders. | Qualitative |
| Green & Colucci, 2020 [33] | | To review traditional healers' and biomedical practitioners' perceptions of collaboration. | Review |
| Herman et al., 2018 [63] | Liberia | To examine the potential for collaboration between traditional and Western medicine to close the mental health treatment gap. | Qualitative |
| Johnson, 2021 [64] | Dominican Republic | To examine attitudes toward traditional healing for mental health disorders among Dominicans. | Mixed, but Qualitative part for traditional healers |
| Kamanga et al., 2019 [65] | Malawi | To explore the knowledge of mental illness and the concept of mental health healing among pastors. | Qualitative |
| Keikelame & Swartz, 2015 [66] | South Africa | Explore traditional healers' perspectives. | Qualitative |
| Khan et al., 2023 [67] | Pakistan | To explore perceptions around treatment options provided by diverse care providers. | Qualitative |
| Khosla & Goel, 2021 [68] | India | To compare the beliefs and attitudes towards traditional healing methods with modern medical treatment. | Qualitative |
| Kokota et al., 2022 [42] | Malawi | To explore the views and experiences of traditional and western medicine practitioners on potential collaboration. | Qualitative |
| Kpobi & Swartz, 2018 [69] | Ghana | To examine the variation of different healers' power perceptions and the relationship between that power and the perceived power of biomedical approaches | Qualitative |
| Kpobi et al., 2024 [70] | Ghana | To develop collaborations between healers and formal health services to optimize available mental health interventions. | Qualitative |
| Lambert et al., 2020 [41] | Ghana | To gather information on beliefs about mental illness and experiences | Qualitative |
| Lampiao et al., 2019 [71] | Malawi | To evaluate the barriers that exist between traditional healers and biomedical practitioners for them to collaborate. | Qualitative |
| Lee, 2015 [72] | Singapore | To summarize the relevance of traditional healing concepts and practices for mental health. | Narrative |
| Longkumer, 2020 [73] | India | To assess the role of traditional healing practices. | Mixed Method |

*(Continued)*

| Author/Year | Country | Aims/objectives | Study Types |
|---|---|---|---|
| Makgabo, 2023 [74] | South Africa | To explore the possibility for such collaboration by investigating case formulation by western-trained clinical psychologists and traditional health practitioners. | Qualitative |
| Maricar, 2018 [75]. | Singapore | To gain a deeper understanding of Malay Muslim healers' roles in treating patients with mental illness. | Qualitative |
| Moghaddam et al., 2015 [76] | America | To demonstrate the value of ritual and traditional healing for urban American Indians/Alaska natives. | Qualitative |
| Molot, 2017 [5] | South Africa | To gain a greater understanding of healers' conceptualizations of mental illness and examine how biomedical practitioners understand and view this parallel healthcare system | Qualitative |
| Moorehead et al., 2015 [77] | America | To report insights from a 4-day gathering of native American healers | Qualitative |
| Mukala Mayoyo et al., 2023 [78] | DR Congo | To identify the current mix of services for mental health care | Qualitative |
| Musyimi et al., 2016 [79] | Kenya | To form dialogue and design collaboration between informal and formal healers. | Qualitative |
| Musyimi et al., 2017 [80] | Kenya | To explore barriers faced by trained informal health practitioners, refer individuals with suspected mental disorders for treatment and potential opportunities to counter these barriers. | Qualitative |
| Nyame et al., 2021 [40] | Ghana | To examine the possibility of forging cooperation at the primary health care level in two geopolitical regions. | Qualitative |
| Olutope, 2020 [81] | Nigeria | To explore the barriers of the African traditional healing system and the implications for the development of indigenous psychotherapy. | Narration does not used a specific design |
| Osafo, 2016 [82] | Ghana | To provide a framework within cooperative linkages between religious leaders. | Qualitative |
| Patterson, 2023 [83] | Tanzania | To examine the ways power and struggle in public authority affect mental health. | Qualitative |
| Pham et al., 2021 [39] | Nepal | To explore how the general public, traditional healers, and biomedical professionals perceive the different types of services and make decisions regarding using one or both types of care. | Qualitative |
| Pullen et al., 2021 [84] | Liberia | To understand the beliefs and beliefs held by traditional healers and utilizers of traditional medicine. | Qualitative |
| Ramakrishnan et al., 2014 [85] | India | To understand the perspectives of the Indian traditional and complementary medicine and allopathic professionals on the influence of religion/ spirituality in health. | Quantitative |
| Shange & Ross, 2022 [86] | South Africa | To explore the beliefs and practices of traditional healers. | qualitative |
| Shields et al., 2016 [44] | India | To explore the origins, use, and outcomes of a collaborative program between faith-based and allopathic mental health practitioners. | Qualitative |
| Solera-Deuchar et al., 2020 [87] | Tanzania | To establish the views of traditional and biomedical practitioners towards collaboration between the two sectors. | Qualitative |
| Soori et al., 2024 [51] | West Africa | To examine the enablers and obstacles to the integration of traditional medicine and mainstream medicine in mental health in West Africa | Review |
| Tafoyan, 2018 [88] | America | To explore the experiences of licensed mental health practitioners who are also traditional Mexican/Mexican American healers in their practice of both approaches. | Qualitative |
| Taruvinga, 2016 [89] | Zimbabwe | To explore the conceptualisation and treatment of mental illness by Zezuru Shona traditional healers. | Qualitative |
| Thomas et al., 2015 [90] | United Arab Emirates | To explore traditional healers' conceptualisations of mental health Issues. | Qualitative |
| van der Watt et al., 2017 [91] | Ghana, Kenya, and Nigeria | To examine the scope of collaborative care for individuals with mental illness as implemented by traditional healers, faith healers, and biomedical practitioners. | qualitative |
| van Niekerk et al., 2014 [92] | South Africa | To identify traditional healers' awareness of occupational therapy, their use of occupations in their treatment. | Mixed Method |
| van Rensburg et al., 2014 [38] | South Africa | To capture the views of some local psychiatrists on referral and collaboration between psychiatrists and religious or spiritual advisers. | Qualitative |

*(Continued)*

**Table 1.** (Continued)

| Author/Year | Country | Aims/objectives | Study Types |
|---|---|---|---|
| Yaro, 2016 [93] | Ghana | To explore the perspectives of different stakeholders about mental health care services provided by traditional healers and the possibility of collaboration with community mental health care | Qualitative |
| (Yaro et al., 2020 [94] | Ghana | To assess beneficiaries' views about the impact of intervention. | Qualitative |

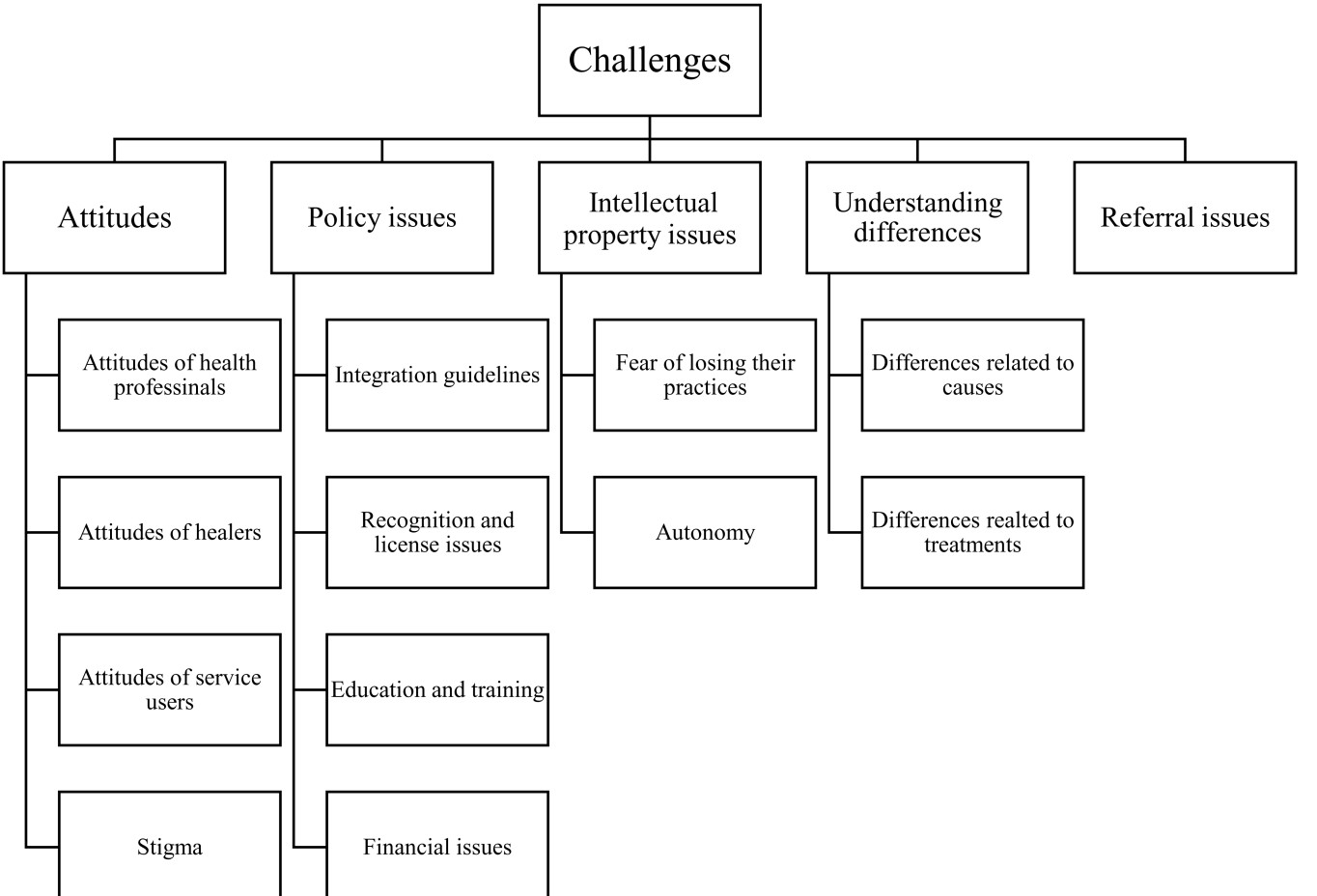

**Fig 2. Shows major themes and subthemes of challenges toward integrating traditional healing approaches with biomedical treatment for mental illness from perspectives of healers.**

believe clinicians would not be willing to collaborate with them [54]. Similarly, the study conducted in South Africa showed healers feel that biomedical practitioners are unwilling to integrate due to their ignorance of traditional healing practices [74]. In addition, healers felt that biomedical providers look down on them, considering them less competent [71]. The study conducted in Kenya showed that traditional healers felt demeaned and stereotyped by clinicians [79]. They felt that Western practitioners had no desire to cooperate or share knowledge with them and they also felt that their practices are always being considered inferior, according to a study conducted in Liberia [63]. The study conducted in Singapore revealed that healers believe that doctors do not acknowledge the veracity of spiritual healing, and as a result, they have no wish to work together [75]. Furthermore, the study conducted in America mentioned the fear

of traditional healers and raised concern that biomedical practitioners does not validate their healing due to a lack of empirical evidence [88].

**Attitudinal challenges of healers:** A total of seven studies showed that traditional healers have their intrinsic issues, which should be considered for effective communication and collaboration with biomedical staff. The study conducted in Uganda indicated those intrinsic barriers, such as competency issues, were mentioned among traditional healers [55]. Internal competition among healers was mentioned as a major challenge to building integration and communication between the two treatment systems, according to the study conducted in Tanzania [83]. Similarly, the presence of fake and unlicensed traditional healers who are not effective in treating people's problems was mentioned as a problem in studies conducted in Nigeria and the United Arab Emirates [81,90]. Another study conducted in Ghana indicated that traditional healers believe that their treatment options are more officious than those of biomedical practitioners; they distrust biomedical treatment methods, do not believe they require recognition or help from biomedicine, and have no desire to associate with biomedical practitioners [69]. The findings of study conducted in the DR Congo were similar and revealed that traditional healers are unwilling to collaborate with biomedical practitioners because they believe biomedical treatment methods contradict the divine world [78]. Faith healers have believe that integrating both approaches is difficult due to a lack of faith or trust in God among some doctors, according to a study conducted in Malawi [65].

**Attitudes of service users:** Some studies (n = 6) report the concerns about traditional healers by service users. Healers believe that some patients and caregivers are reluctant to use clinical services due to the fear of interfering with God's order [34]. Similarly, according to a study conducted in India, patients and caregivers report discomfort if healers refer them to hospitals [85]. The study conducted in the USA showed that service users' feel ashamed to discuss, try to hide, or not be as open about practicing traditional medicine [88]. The study conducted in Singapore noted the consideration of patients' views before starting collaborative practices [72]. Patients' fear of injections provided by biomedical practitioners due to side effects was indicated, according to the study conducted in Liberia [84]. The study conducted in Tanzania showed that some biomedical proponents or service users' consider healers' treatment unscientific, dangerous, and an obstacle to helping people with mental illness [83].

**Stigma:** There are some studies that showed traditional healers' concerns related to stigma that challenged open communication with biomedical professionals and service users. The study conducted in Ghana showed stigmatization related to mental illness [58,82]. Similarly, another study conducted in Ghana showed that sometimes families abandon patients from the services due to stigma related to mental illness, while in Pakistan the presence of stigma toward the activities of traditional healers and people who took their services were revealed [67]. Stigma-related issues are also mentioned as one problem for open and collaborative treatment for mental illness, according to a study conducted in South Africa [38]. Likewise, the study conducted in Kenya showed that, most often, patients have negative and stigmatized beliefs related to mental illness, which are commonly associated with fear of being mishandled or bewitched [80].

**Policy issues:** According to the included studies, policy and direction problems are often cited as challenges for traditional healers when integrating two therapeutic approaches. The majority of the included articles showed the absence of integration guidelines, limitations in education and training, the lack of recognition and license for traditional healers, and financial concerns as priority problems that hinder collaboration.

**Integration guidelines:** Among the total number of included studies, twelve focused on guideline-related challenges. The study conducted in Ghana showed the absence of clear communication between the two systems and tension and disagreement between them [82]. Another study conducted in Ghana indicated the need for political commitment for effective collaboration [93]. Traditional healers mentioned the absence of any means of agreement to communicate with biomedical practitioners, according to a study conducted in South Africa [66]. Similarly, the absence of consensus on integration and the lack of an open dialogue to foster mutual understanding were identified as challenges for integration, according to the study conducted in Nepal [39]. The studies conducted in the United Arab Emirates and India indicated

a lack of regulatory processes and the absence of clear guidelines for effective communication [56,90]. Likewise, the absence of consensus on integration and referral were mentioned as problems in a study conducted in Uganda [52]; lack of clear evidence about specific types of disease to be integrated was mentioned as a challenge in a study conducted in Singapore; and, lack of a platform or mechanism for communication and dialogue to foster effective cooperation is noted in other studies [44,71,81]. Furthermore, the study conducted in Kuwait showed that faith healers have concerns due to a lack of a clear job description for each treatment approach before considering integration [57].

**Recognition and license issues:** Limitations in recognizing and licensing traditional healers as professionals were reported I five (n = 5) articles. The study conducted in Kenya showed that lack of recognition for traditional healers enforces their desire to perform secretive practices, and complained that some healers have been prohibited from practicing their traditional healing process [53]. In another study conducted in the United Arab Emirates, the presence of unlicensed healers was mentioned as a risk of communication between the two treatment approaches [90]. Similarly, some traditional healers mentioned that they are not allowed to practice their healing process due to government restrictions [83]. The study conducted in America mentioned a lack of recognition for traditional healing, while faith healing practices are considered illegal in Kuwait [57].

**Education and training:** Education and training gaps were another policy-related challenge mentioned in 10 (n = 10) studies. A study conducted in Nigeria showed that excluding traditional medicine from health sciences students' curricula was challenging for good understanding and effective communication [81]. Similarly, a study in India noted the absence of spirituality in the curriculum [85]. The training gap and healers' differences with biomedical staff were identified as challenges for effective communication among traditional healers in studies conducted in Ghana [61,93]. The studies conducted in Nepal and South Africa showed that education problems and a lack of understanding between the two modes of treatment were challenges for integration [39,86]. Traditional healers did not know the procedures they have in hospitals or the medications they use, according to a study conducted in Tanzania [87]. Similarly, the education gap and mutual understanding issues are mentioned as challenges according to the study conducted in South Africa, and the study conducted in Singapore showed the presence of their own master in healing and indicated the need for education to facilitate mutual communication [75]. The study conducted in the Dominican Republic showed that traditional healers lack education and training due to the absence of traditional healing in health care system curriculum [64].

**Financial issues:** Some studies reported financial barriers to effective integration of traditional healers and biomedical professionals. The study conducted in Liberia showed that people with mental health problems might face challenges in affording medication [84]. In addition, the high cost of modern treatments is mentioned in studies conducted in India and Zimbabwe [68,89]. Similarly, logistic rearrangements like treatment procedure location, office, and medication are mentioned as challenges that governments should consider for effective integration [40,74]. The study in Malawi showed a lack of commitment and financial support from the government for the integration [42].

**Intellectual property issues:** Intellectual property issues (fear of losing their healing practices and autonomy), were noted as barriers to considering the integration of both approaches for mental illness in 14 studies.

**Fear of losing their practices:** The study conducted in Uganda showed traditional healers fear of losing their position by biomedical practitioners [55]. Similarly, they are concerned that modern medicine interferes with their profession and may inhibit fasting and related religious activities [58]. The study conducted in Ghana also indicated that the fear of losing their position as a major threat mentioned by traditional healers; they believed that if they consider integration, biomedical professionals may dominate their practices [69]. Some studies show traditional healers' concerns that modern health professionals may steal their skills and practices without recognizing them [42,91]. They fear that their medicine will lose value if they integrate with Western medicine. This is explained as: "They will kill it with their chemicals, turn it weak, and cause it to be unable to work as strong as it should be" [86]. Fear of being used and having medicinal knowledge and practices is also traditional healers' concern for integration, as mentioned in the included studies [86,93]. According to a

 

study conducted in South Africa, the protection of intellectual property and accountability using formal agreements were stressed by traditional healers before considering integrative works [66].

**Autonomy:** Traditional healers mentioned autonomy issues as challenging for integrative work. The study conducted in Kenya showed that healers have concerns that integration is only defined from the perspective of biomedical practitioners, which might threaten their status, economy, and freedom [53]. They also mentioned the power dynamic between the two and the unfair competition of doctors [52,89]. The study conducted in Ghana showed control, autonomy, and power in biomedical mental health practitioners as challenges from the healers' perspectives [60]. Similarly, the study conducted in Kuwait indicated that each other's superiority, thinking that their approach is more effective, and not being convinced of the other side's effectiveness were challenges for collaboration [57]. Likewise, the study conducted in Nigeria showed poor protection of knowledge and intellectual property rights, or autonomy, as a significant concern for integrating traditional medicine with modern treatment [81].

**Challenges related to conceptual understanding differences:** The studies showed that traditional healers have significant concerns due to their basic conceptual understanding of differences in mental illness causes and treatment approaches.

**Understanding differences related to the cause of mental illness:** Challenges due to understanding differences in the cause of mental illness are mentioned in eight studies. The study conducted in Uganda showed that the cause of mental illness is believed to be spiritual [55]. In another study, attributing supernatural causes for mental illness was given as a reason for healers' communication difficulties with doctors [82]. The studies conducted in South Africa and India revealed that disease conceptualization and understanding differences were challenges for collaborative work between the two treatment approaches [73,74]. Healers conceptualization of problems as supernatural rather than biomedical and their belief that biomedical methods cannot address supernatural causes are reported as challenges for effective collaboration, according to other studies conducted in India [44,68]. The study conducted in Ghana, Nigeria, and Kenya showed that differences in religious systems between healers were mentioned since it is challenging to work together with different ideologies [91]. Traditional healers believe that most mental illnesses have non-biological origins and should, therefore, be addressed by traditional medicine, according to the study conducted in the DR Congo [78].

**Understanding differences related to treatments for mental illness:** Challenges related to treatment differences are indicated in eight studies. Traditional healers believed that their approach was the only effective treatment for mental illness, and they felt that clinicians were poorly paced in improving spiritual origins, as indicated in a study conducted in Uganda [55]. The study conducted in Liberia showed belief in God and spirituality as means of treatment for mental illness [84]. Another study conducted in Liberia noted the presence of specific diseases treated explicitly by traditional healers [63]. Traditional healers cited that biomedicine could not tackle spiritual issues, and some of them believe mental illness does not need modern treatment [91]. A study conducted in Tanzania showed healers' beliefs that patients possessed by a *jinn* (a spirit) or those who had been bewitched needed only traditional treatments [87]. Similarly, the study conducted in South Africa showed traditional healers' beliefs that allopathic medicine cannot treat and remove evil spirits or misfortune, and they also believed that allopathic treatment methods are temporary [92]. The presence of some illnesses that biomedical doctors could not cure and medications given by doctors cannot heal people as quickly as their healing was indicated in South Africa [5].

**Referral issues:** Of the included studies, ten showed the healers' identified referral procedures and systems as roadblocks to effective communication between the two methods. The study done in South Africa found that one of the challenges is that allopathic practitioners do not provide back referrals, and they often advise patients not to return for traditional health consultation [92]. Similarly, the Liberia study revealed that biomedical personnel did not refer patients back to them [66]. According to a study done in Nepal, healers have tried sending their patients to health centers, but doctors didn't refer to traditional healers [39]. Healers believe they are solely required to refer patients to Western medicine in a one-way manner, and this is seen as a significant integration barrier [5,89]. In addition, traditional healers believe that

biomedical practitioners are unwilling to recommend patients to them because they feel traditional healers lack understanding [60]. The study conducted in Ethiopia identified absence of referrals from biomedical staff, and healers' concern that biomedical practitioners hold negative attitudes toward patients they refer to hospitals [34]. In addition, healers believe that referring patients to doctors may conflict with their belief system, and patients may refuse to go to the hospital [79]. The study conducted in Tanzania showed that because of the absence of formal referrals, healers are facing challenges in communicating with biomedical professionals [87]. According to a study conducted in Liberia, traditional healers are also frustrated due to the lack of reciprocal referrals from biomedical practitioners [63].

**Opportunities for integration from the healers' perspectives:** Studies have focused more on the challenges than the opportunities of integrating traditional healing with biomedical modalities for mental illness. The World Health Organization's (WHO) recognition of traditional medicine, treatment effectiveness, and willingness to work together are reported by healers as opportunities for the cooperative work of both therapeutic modalities [50,56,72,95]

According to the study conducted in India, the perceived effectiveness and alignment of traditional healing practices with local cultural beliefs are mentioned as enforcing factors for communication [56]. Studies also indicated that both approaches were effective methods to improve the symptoms of mental illness [76,82]. The study conducted in Nepal showed some understanding between both fields as a viable option for integration [39], with traditional healing practices considered affordable and practical for the treatment of mental illness treatment [89]. Similarly, the study conducted in Ghana reported that a person who combines spiritual exercise with medicines from nurses recovers faster [94].

Additional facilitators for collaboration included the WHO's recognition and providing a positive environment for traditional medicine. Similarly, the integration of traditional healing practices into modern healthcare systems is gaining recognition [59]. Even most of the practitioners recognise that patients can benefit from combining both practices [33]. This recognition facilitates training, knowledge sharing, and a mutual referral system [94].

Furthermore, the willingness of traditional healers to collaborate is mentioned as an important opportunity for integration. A study conducted in Liberia showed that some healers are open to learning about modern approaches [84], and they have a desire to collaborate with biomedical practitioners [50]. Even some healers have the experience of referring their patients to biomedical practitioners [34], and referral and training were believed to be the preferred forms of collaboration [42]. Healers interest in collaboration with biomedical mental health care providers was mentioned as facilitator for collaboration, according to the study conducted in Kenya [62]. Mutual respect, honest communication, and the need for awareness of cultural differences were noted as values for integrative treatment of mental health problems [77]. The study conducted in Kuwait showed that most participants were willing to collaborate with psychiatrists. Their views are explained as "we want to collaborate and learn from them to enhance our knowledge [57]." Some healers believe that collaboration is important to get recognition and financial support for their work [57]. The study conducted in Ghana showed successful collaborations through mutually respectful interpersonal relationships, support from the health system, and access to community resources [70].

## Discussion

Traditional healers provide an important treatment option for people with mental illness, particularly in low-income countries, and most people seek traditional healers because they are believed to provide culturally and socially accepted care [96–99]. The World Health Organization's (WHO) 2013–2020 Mental Health Action Plan also recommends that government health programs incorporate traditional healers as treatment resources [95]. Despite the importance of integrating traditional healing with biomedical care to improve the outcomes of people with mental health conditions, there is still limited communication between the two treatment approaches. This scoping review summarizes the contemporary evidence on challenges and opportunities of integrating traditional healing with biomedical care for mental illness from healers' perspectives, including globally available published and unpublished studies. The scoping review is preferable to addressing broad literature related to problems since it allows different types of studies without focusing on their qualities [100].

In addition, the scoping review is an appropriate technique to identify the knowledge gap and summarize critical factors related to the review topic [100,101].

## Challenges of integrating traditional healing with biomedical treatment

Although collaboration can lead to more holistic mental health and improve treatment processes and outcomes for individuals with mental illness symptoms, this review found several challenges that prevent biomedical practitioners and traditional healers from effectively integrating their practices.

### Attitudinal challenges

Major prohibiting factors documented in a significant number of the included publications are the negative attitudes of the health care practitioners, healers' themselves, and the service users. When there is no mutual trust and understanding, cooperative work remains problematic. Healers typically believe that medical experts view their methods as unhygienic, which makes them reluctant to integrate [55]. Suspiciousness and a negative attitude can lead to continued criticism, disrespect, and ignorance [39–41,61]. Commonly, negative attitudes exist among biomedical practitioners who disregard healers, consider their techniques ineffective, and often regard them as witches [79]. This influences healers to undermine themselves and to proceed with secret practices rather than promoting themselves. Studies stressed that attitudinal differences are also an area of conflict and tension between traditional medicine and biomedical professionals [51,82,91]. Health professionals priority for scientific evidence, and mistrust of healers prevents them from working together [102]. In addition, some faith healers also regard biological treatment as a practice apart from God and concentrate on prayer and God's will for treating mental illness; faith doesn't encourage cooperation and trust between two systems [65]. Care providers and service users' attitudes are vital for integrative activity. Positive attitudes and understanding both treatment systems' goals, advantages, and consequences are essential for patients and their caregivers. Because they feared going against God's will, the result suggested that more spiritual patients and caregivers were reluctant to obtain biomedical care [34]. This shows that it is challenging without an awareness of and comfort with service users [85]. Likewise, families leave their patients without proper treatment due to stigma; there is a direct link between treatment choices made by service users and the social stigma associated with mental illness [61,67]. Furthermore, research backs up the notion that stigmatizing views cause patients and caregivers to procrastinate and not participate in joint efforts [103]. Traditional healers also faced challenges from the stigmatization of mental illness, the services provided by them, and the abonnement of clients by family caregivers [58]. Similarly, another study showed that stigmatizing beliefs and unfavourable views are common obstacles for patients to receive the treatment they need [104–106].

### Challenges related to policy

Without a clear policy, the two treatment modalities may not integrate as needed, which could have a significant influence on mental health, especially in low-income nations with huge treatment gaps. As a result, a well-defined policy can offer answers and guidance for integration guidelines, training gaps, financial allocation concerns, traditional healers' recognition and licensing issues, and others that are frequently mentioned as challenges. From the healers' perspective, challenges related to policy included either the absence of clear direction for integration [39,82] or the non-functionality of available policies [57]. The requirement for precise standards, including job descriptions for two treatment modalities, was the most significant issue that healers noted as a government policy issue [56,64,90]. It would be impossible to encourage joint action among healers and health care professionals without policy-backed guidelines. Traditional healers believe guidelines should incorporate specific alignment categories [72]. Traditional healers' lack of acknowledgment and licensing concerns are another crucial issue that must be addressed explicitly to establish acceptable collaboration because unlicensed healers [90]. Without clear guidance, even certain government personnel were reluctant and considered traditional healing practices illegal [57,83]. Another urgent policy issue relates to mutual education and training issues

since some governments have neglected Indigenous treatment [81,85]. Treating patients with different values, beliefs, and behaviours without mutual understanding is challenging. These include adjusting care delivery to the patients' social, cultural, and linguistic demands. Similarly, without training, healers would not comprehend the processes and drugs used in medical institutions [87]. Furthermore, governments should consider the financial concerns of service users, including the costs of biomedical treatment, while taking an integrative approach. This is because logistics, such as medication and shared offices, are problems mentioned from healers' perspectives [40,68,74,84,89]. This finding aligns with a review study in West African nations, which identified policy and implementation concerns as a primary impediment to integrating two treatment modalities [51]. The review is supported by the study conducted on traditional birth attendants, which showed a lack of understanding and financial issues like equipment as challenges of integration with biomedical treatment approaches [107]. Misunderstandings regarding each other's practices were mentioned as barriers to effective communication [108].

## Intellectual property issues

Intellectual property conflict is another major area that needs solutions for the integrative work of two treatment approaches because there are complexities regarding their professional existence and autonomy while collaborating two systems with different power and healing approaches. Autonomy issues and fear of losing their professions are merged as risks in different included studies [52,53,55,58,69,89]. For example, healers have a fear of their medication being destroyed with biomedical professionals' chemicals, turning it weak and making it unable to work as well as it should, and this forces the clear consensus on protection of intellectual property and accountability issues [66,81]. There is other supporting evidence for this finding; the lack of protection for traditional knowledge and their intellectual property rights is a concern [109]. The power imbalance between biomedical and traditional practitioners is mentioned as aa obstacle to integration [110]. Indigenous knowledge is at risk without protection of intellectual property [111].

## Understanding differences

Another major obstacle preventing healers from working collaboratively are differing conceptualizations and understandings of the causes and treatment mechanisms for mental illness. This is because, in traditional healers' communities, the majority of perceived causes are supernatural forces that have non-biological origins, such as spirits, Jinn, and witchcraft [44,55,61,68,82,87]. As a result, they believe that spirituality, or belief in God, and practicing supernatural treatment are the only means of treatment [84,91]. This suggests a lack of coherent and mutually acceptable understandings of the management and causality approaches to mental illness, which prevents the possibility of cooperation between the two therapy modalities. Thus, treatment options can be unpredictable since there are differences in disease definition and diagnosis methodologies. Conceptualization may not differ only between biomedical doctors and traditional healers but also between healers themselves [33,51,112]. Proponents and service users' understanding are also strongly impacted since they largely follow the healers' ideas and resist other means of understanding and treatment preferences [113]. The differences in health beliefs and misunderstandings regarding each other's treatment approaches are reported as challenges for effective communication [108]. Similarly, religious belief's effect on health-seeking behaviour and its influence on integrating two health care systems [114].

## Referral system

Referral processes are another important consideration, as they serve as a significant channel for sharing patients and responsibilities between the two approaches. One of the most frequently identified issue in this review is that healers are not receiving reciprocal referrals from biomedical professionals, even when they are sought [34,39,66,89]. Healers expressed concerns that biomedical care providers have a negative attitude towards traditional healers and advise patients not to return for traditional health consultation, in addition to the lack of back referrals from allopathic practitioners

[34,79,92]. Due to this, healers believe they are considered as having no knowledge, which leads them to abstain from collaborative work [60,79]. In order to foster productive collaboration, consensus should be reached regarding referral standards and the creation of referral practices. Effective communication, respect for one another's practices, and shared patients may be necessary for this to occur. As most healers do not have formal methods of communication with biomedical professionals, mutually relevant and developed referral systems could facilitate the reciprocal referral procedures [87].

## Opportunities for integrating traditional healing with biomedical treatment

Although studies gave limited focus to identifying opportunities for integrating traditional health approaches with biomedical treatment from the perspectives of healers, the WHO's recognition of traditional medicine, the effectiveness of traditional healing for mental illness, and the openness of some healers to cooperation are important areas of potential collaboration. The WHO is urging holistic care for mental illness to overcome large treatment gaps, particularly in low-income countries. The WHO's recommendation to consider traditional medicine is an important reason for integration to be considered [72]. In addition, the alignment of traditional healing approaches with local cultural beliefs is another important factor for collaboration between the two approaches to occur [56,59,82]. Studies have shown the effectiveness of collaborative treatment for mental illness, but without the commitment of governments and other stakeholders, policy, training and education, and referral issues will continue to hinder integration [51,94]. Well-established training and education can foster systemic integration, decision-making, and service delivery for the two treatment approaches [51]. Some traditional healers desire to learn about biomedical treatment processes [84], and have an openness to collaboration [50]. Mutual respect and the need for awareness of cultural differences are good incentives for integrative thinking [77]. Biomedical professionals working with healers can also potentially improve their knowledge, and service users' can benefit from the openness and effective referral pathways between healers and biomedical partitioners [42,94,115–117].

## Limitations

The review incorporated published studies from five databases and unpublished studies from different sources to widen the available evidence, but it is possible that additional findings could have been obtained from other databases. In addition, the review may have overlooked studies in screening because some database outputs are handled using the title and abstract screening processes. Furthermore, this review is limited by the omission of publications that are not written in English.

## Conclusions

The WHO recommends consideration of evidence-based traditional medicines to close the treatment gap for mental illness, especially in low-income countries. This scoping review identified challenges to effective collaboration between traditional and biomedical practices, including negative attitudes, policy issues, intellectual property, disease understanding differences, and referral issues. In this review, the WHO's recognition, the effectiveness of traditional healing, and willingness of healers to integrate were identified as enabling factors for integration. Although, understandably, challenges related to integration vary based on the sociocultural perspectives of different countries, policymakers and other stakeholders. In addition, limited evidence, particularly related to opportunities for integrating the two treatment approaches, indicates that future research should focus on diverse sociocultural contexts to widen the evidence on challenges and opportunities for effective cooperation.

## Acknowledgments

The first author of this review is a recipient of an Australian Government research training scholarship through the University of New England. In addition, we would like to thank the Dixson library staff at the University of New England for their support with the search strategy.

## Author contributions

**Conceptualization:** Alemayehu Molla Wollie.

**Formal analysis:** Alemayehu Molla Wollie.

**Investigation:** Alemayehu Molla Wollie, Kim Usher, Kylie Rice, Md Shahidul Islam.

**Methodology:** Alemayehu Molla Wollie, Kim Usher, Kylie Rice, Md Shahidul Islam.

**Supervision:** Kim Usher, Kylie Rice, Md Shahidul Islam.

**Validation:** Kim Usher, Kylie Rice, Md Shahidul Islam.

**Visualization:** Alemayehu Molla Wollie.

**Writing – original draft:** Alemayehu Molla Wollie.

**Writing – review & editing:** Alemayehu Molla Wollie, Kim Usher, Kylie Rice, Md Shahidul Islam.

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
