## [Decision Letter · Decision Letter 0]

2 Apr 2025

PONE-D-24-53011Challenges and opportunities for integrating biomedical care with traditional healing for mental illness: A scoping review from healers’ perspectivesPLOS ONE

Dear Dr.  Wollie,

Thank you for submitting your manuscript to PLOS ONE. After careful consideration, we feel that it has merit but does not fully meet PLOS ONE’s publication criteria as it currently stands. Therefore, we invite you to submit a revised version of the manuscript that addresses the points raised during the review process.

We look forward to receiving your revised manuscript.

Kind regards,

Godwin Upoki Anywar, BSc, Msc, PhD

Academic Editor

PLOS ONE

Journal Requirements:

3. Please include captions for your Supporting Information files at the end of your manuscript, and update any in-text citations to match accordingly. Please see our Supporting Information guidelines for more information: http://journals.plos.org/plosone/s/supporting-information .

4. As required by our policy on Data Availability, please ensure your manuscript or supplementary information includes the following:

Reviewers' comments:

Reviewer's Responses to Questions

**Comments to the Author**

1. Is the manuscript technically sound, and do the data support the conclusions?

Reviewer #1: Yes

Reviewer #2: Yes

2. Has the statistical analysis been performed appropriately and rigorously? 

Reviewer #1: N/A

Reviewer #2: Yes

3. Have the authors made all data underlying the findings in their manuscript fully available?

Reviewer #1: Yes

Reviewer #2: No

4. Is the manuscript presented in an intelligible fashion and written in standard English?

Reviewer #1: Yes

Reviewer #2: Yes

5. Review Comments to the Author

Reviewer #1: The conclusion in the abstract is a bit confusing. Here you are integrating people (traditional healers) with biomedical approaches. Is this real? Why not integrate healers practices with biomedical approaches?

Background: Traditional healing is defined in a very narrow way by only one reference. I suggest you get more references that may define traditional healing more broadly.

Look at the comments on the few following highlights in the manuscript for your reflection

Otherwise, I have enjoyed reading the outcome of the manuscript.

Reviewer #2: Title - Integrating Biomedical Care with Traditional Healing vs Integrating Traditional Treatment Approaches with Biomedical Treatment, which sounds and is described differently in the document. Does it have differences? In addition, integration and collaboration terms are different in health systems, therefore, it needs operation definition in this research.

Background:

- it is good to add existing evidence related to integrated services effectiveness, the challenges and its opportunities and consider the word limit of the journal. as, there is no compiled evidence on the challenges and opportunities related to integrating traditional healing practices with modern mental health care from the healers’ perspectives. This statement seems very general as the evidence which may be not adequate and your research gap filling should be clearly addressed.

- Do you think the global context has similar integration challenges and opportunities? I recommend addressing this issue in different context such as European context, Asian , Africa etc as background content . then, it can be help during the discussion and thematic result writing process.

Methodology:

- What was the reason for the search was limited to January 2014 and June 30, 2024?

- Was the list of synonyms or alternative terms developed for the initial keywords for the search strategy?

- Was it done manually? After removing duplicates and filtering important articles, the data were extracted using the prepared data extraction form

- Would describe clearly about thematizing process including the coding process example was it deductive or inductive, was it coded only by one person? Use reference for the thematic analysis as there are different steps and guides of thematic framework analysis.

- Do you use any tool to assure the quality of the included articles?

- Describe the analysis process to assure the trustworthiness meaning the validity of quality of the research data. Example, good to include the annex of Nvivo exported codes indexation and make clear if the coding process was done by independent reviewers.

Result:

- Under the review of studies. It is described that three review articles are included. It is not clear that if all articles are included in your study, why the existing review studies included that means the single articles finding extracted data, again be extracted their finding indirectly from the other reviews citation(if included). In addition, if that is the case, the main significance of this review in comparison to previously published reviews needs to be clearly justified. Could you elaborate on this point to clarify the unique contribution and added value of your review?

- Attitude challenges related to health professionals …“Emenet,” what is does mean ?

- Attitude challenges related to health professionals line 20 - biomedical staff Vs , under Attitudinal challenges of healers line 2 - western practitioners ….. would you make it consistent through all the documents in using such terms?

- The result writing flows to make it easy for readers, I prefer to make it starting from developed to developing or something else flow. some paragraphs start to state the result findings from Africa to UAE and then to Africa. if the statements are not interconnected, I prefer stating the findings with similar contexts.

- Attitudes of service users, line 2 - Some patients and caregivers are reluctant…. Are studies among such participants (patients and caregivers) included in your review?

- Policy issues – references to the articles are not included.

Discussion

- The repetition of similar subtopics in the result and in the discussion including repeating results of some paragraphs in the discussion are already in the way they are written in the result … this makes it boring for readers and better to consider a revision.

6. PLOS authors have the option to publish the peer review history of their article (what does this mean? ). If published, this will include your full peer review and any attached files.

**Do you want your identity to be public for this peer review?** For information about this choice, including consent withdrawal, please see our Privacy Policy .

Reviewer #1: **Yes: ** Yahaya Hills Kagali Sekagya

Reviewer #2: No

---

## [Author Response · Author response to Decision Letter 0]

23 Apr 2025

Responses to reviewers’ comments

PONE-D-24-53011 Challenges and opportunities for integrating biomedical care with traditional healing for mental illness: A scoping review from healers’ perspectives

Dear editor,

Thank you very much for allowing us to revise our manuscript. We found all of the comments of the reviewers were very important to improving the quality of the manuscript, and we have tried to revise our manuscript accordingly. We hope this version is suitable for publication.

Reviewer #1

Comment: The conclusion in the abstract is a bit confusing. Here you are integrating people (traditional healers) with biomedical approaches. Is this real? Why not integrate healers practices with biomedical approaches?

Response: Thank you very much for giving your important time to review this manuscript. We have revised this as ‘integrative work between traditional healing approaches and biomedical service presents an opportunity to assist in addressing the treatment gap for mental illness.’

Comment: Background: Traditional healing is defined in a very narrow way by only one reference. I suggest you get more references that may define traditional healing more broadly.

Response: Thanks. We have included a broad definition with an additional reference (paragraph 1).

Comment: Look at the comments on the few following highlights in the manuscript for your reflection. Otherwise, I have enjoyed reading the outcome of the manuscript.

Response: Thanks a lot. We have seen your comments and tried to revise document accordingly (please see the track change).

Reviewer #2

Dear reviewer,

We really appreciate you giving important comments, and we have tried to revise this manuscript based on your feedback.

Comment: Title - Integrating Biomedical Care with Traditional Healing vs Integrating Traditional Treatment Approaches with Biomedical Treatment, which sounds and is described differently in the document. Does it have differences? In addition, integration and collaboration terms are different in health systems, therefore, it needs operation definition in this research.

Response:

Actually, we believe the two concepts are similar, but for clarification, we have modified our title as ‘Challenges and Opportunities for Integrating Traditional Healing Approaches with Biomedical Care for Mental Illness.’ In addition, although literature commonly uses the words ‘integration’ and ‘collaboration’ interchangeably, there is literally a conceptual difference between the two words. For example, integration is incorporating or combining different services to create a coordinated approach to improve the effectiveness and quality of health care, but collaboration may be simply communication, relation, or client sharing without strong, well-defined system integration. Therefore, to provide holistic care to service users, we believe service and system integration from different perspectives is important, and that is why we used the word ‘integration’, which is also frequently mentioned in incorporated studies.

Background:

Comment: it is good to add existing evidence related to integrated services effectiveness, the challenges and its opportunities and consider the word limit of the journal. as, there is no compiled evidence on the challenges and opportunities related to integrating traditional healing practices with modern mental health care from the healers’ perspectives. This statement seems very general as the evidence which may be not adequate and your research gap filling should be clearly addressed.

Reponses: Thank you so much for giving this important comment. We have tried to include some evidence regarding this (Page 2, last paragraph).

Comment: Do you think the global context has similar integration challenges and opportunities? I recommend addressing this issue in different context such as European context, Asian, Africa etc as background content . then, it can be help during the discussion and thematic result writing process.

Response: We have tried to incorporate some concepts in the revised version (page 3).

Methodology:

Comment: What was the reason for the search was limited to January 2014 and June 30, 2024?

Response: Thanks. The main reason for including studies published within the last 10 years was to summarize contemporary evidence (the WHO traditional medicine strategy was also updated in 2014). We have provided this rationale in the revised version.

Comment: Was the list of synonyms or alternative terms developed for the initial keywords for the search strategy?

Response: Yes, we did detailed searching using different synonyms for keywords, and we have mentioned this in the revised version.

Comment: Was it done manually? After removing duplicates and filtering important articles, the data were extracted using the prepared data extraction form

Response: Duplicate removal and related reference management were done using EndNote version 20, and we have indicated this in the revised manuscript.

Comment: Would describe clearly about thematizing process including the coding process example was it deductive or inductive, was it coded only by one person? Use reference for the thematic analysis as there are different steps and guides of thematic framework analysis.

Response: Thank you very much. This was an inductive approach. After data extraction, frequent reading was conducted to become familiar with basic concepts. The primary author conducted initial coding and theme generation, which was confirmed by other authors through regular meetings and sharing screens. We have revised our manuscript accordingly and also incorporated a citation for this (page 5).

Comment: Do you use any tool to assure the quality of the included articles?

Response: We haven’t used quality assessment for included articles since, for scoping review, it is not mandatory to do a critical appraisal for each article. But Asksey and O’Malley’s framework and the Preferred Reporting Items for Systematic Review and Meta-Analysis Extension for Scoping Review (PRISM-ScR) guidelines were followed to conduct this scoping review.

Comment: Describe the analysis process to assure the trustworthiness meaning the validity of quality of the research data. Example, good to include the annex of Nvivo exported codes indexation and make clear if the coding process was done by independent reviewers.

Response: As we have explained above, overall analysis was confirmed by continuous meetings of authors sharing NVivo software together, and concepts from figure 2 were direct outputs of NVivo software

Result:

Comment: Under the review of studies. It is described that three review articles are included. It is not clear that if all articles are included in your study, why the existing review studies included that means the single articles finding extracted data, again be extracted their finding indirectly from the other reviews citation (if included). In addition, if that is the case, the main significance of this review in comparison to previously published reviews needs to be clearly justified. Could you elaborate on this point to clarify the unique contribution and added value of your review?

Response: Thank you very much for this critical comment. Three review articles included in this paper were not direct reflections of our study. Their objective was not to summarize the challenges and opportunities of integration of traditional healing approaches with biomedical care from the perspectives of traditional healers, but we found some evidence indirectly regarding this issue while reading overall documents. Among the three included review articles, one was conducted in a single country, Ghana; the second was conducted in a few West African countries; and the third was conducted in seven countries from resource-limited areas. In addition, not all studies of included review articles were eligible for our study, but there were few single studies that were also part of our scoping review. Since our study is a scoping review that incorporated 54 articles, we believe that rather than excluding articles, it is good to include contemporary literature related to problem to show inclusive evidence.

Comment: Attitude challenges related to health professionals …“Emenet,” what is does mean ?

Response: This was a local name directly taken from the included study, and its meaning is holy mud. We have included its English meaning before this word.

Comment: Attitude challenges related to health professionals line 20 - biomedical staff Vs , under Attitudinal challenges of healers line 2 - western practitioners ….. would you make it consistent through all the documents in using such terms?

Response: Thanks. This has been corrected in the revised version.

Comment: The result writing flows to make it easy for readers, I prefer to make it starting from developed to developing or something else flow. some paragraphs start to state the result findings from Africa to UAE and then to Africa. if the statements are not interconnected, I prefer stating the findings with similar contexts.

Response: Thank you very much for this comment. The overall flow of the result has been constructed based on conceptual similarities (themes), which is challenging to rearrange from developed countries to developing countries. However, we have clearly specified the country of origin to facilitate interpretation of the identified papers.

Comment: Attitudes of service users, line 2 - Some patients and caregivers are reluctant…. Are studies among such participants (patients and caregivers) included in your review?

Response: Thanks. This expression was from the perspectives of traditional healers, and we have tried to modify this statement for clarity.

Comment: Policy issues – references to the articles are not included.

Response: Thanks. Under policy issues, we have sub-themes (integration guidelines, recognition and license, education/training, and finance), and citations are incorporated under each sub-theme (page 10-12).

Discussion

Comment: The repetition of similar subtopics in the result and in the discussion including repeating results of some paragraphs in the discussion are already in the way they are written in the result … this makes it boring for readers and better to consider a revision.

Response: Thank you so much. We haven’t repeated subtopics (subthemes) from the discussion section, but we used major themes just to interpret our findings with related other concepts. In addition, the majority of concepts from the discussion were interpretations of our results, not direct repetitions. However, we appreciate your feedback and we have tried to review the manuscript again in order to ensure it is streamlined and not repetitive.

---

## [Decision Letter · Decision Letter 1]

12 May 2025

Challenges and opportunities for integrating traditional healing approaches with biomedical care for mental illness: A scoping review from healers’ perspectives

PONE-D-24-53011R1

Dear Dr. Wollie,

We’re pleased to inform you that your manuscript has been judged scientifically suitable for publication and will be formally accepted for publication once it meets all outstanding technical requirements.

Kind regards,

Godwin Upoki Anywar, BSc, Msc, PhD

Academic Editor

PLOS ONE

Additional Editor Comments (optional):

Reviewers' comments:

Reviewer's Responses to Questions

**Comments to the Author**

1. If the authors have adequately addressed your comments raised in a previous round of review and you feel that this manuscript is now acceptable for publication, you may indicate that here to bypass the “Comments to the Author” section, enter your conflict of interest statement in the “Confidential to Editor” section, and submit your "Accept" recommendation.

Reviewer #1: All comments have been addressed

Reviewer #2: (No Response)

2. Is the manuscript technically sound, and do the data support the conclusions?

Reviewer #1: Yes

Reviewer #2: Yes

3. Has the statistical analysis been performed appropriately and rigorously? 

Reviewer #1: N/A

Reviewer #2: Yes

4. Have the authors made all data underlying the findings in their manuscript fully available?

Reviewer #1: Yes

Reviewer #2: No

5. Is the manuscript presented in an intelligible fashion and written in standard English?

Reviewer #1: Yes

Reviewer #2: Yes

6. Review Comments to the Author

Reviewer #1: (No Response)

Reviewer #2: The comments have been well addressed. However, I recommended adding the exported codes from the NVivo software as an annex or supplementary document to the manuscript, but they have still not been included. Therefore, even though Figure 2 provides a summary of the subthemes, I prefer also that you have to include the exported codebook summary of the indexed data from the NVivo software to be sent to us for review before recommending for publication.

7. PLOS authors have the option to publish the peer review history of their article (what does this mean? ). If published, this will include your full peer review and any attached files.

**Do you want your identity to be public for this peer review?** For information about this choice, including consent withdrawal, please see our Privacy Policy .

Reviewer #1: **Yes: ** Dr. Yahaya Hills Kagali Sekagya (MD., Ph.D.)

Reviewer #2: No

---

## [Editor Report · Acceptance letter]

PONE-D-24-53011R1

PLOS ONE

Dear Dr. Wollie,

I'm pleased to inform you that your manuscript has been deemed suitable for publication in PLOS ONE. Congratulations! Your manuscript is now being handed over to our production team.

Kind regards,

on behalf of

Dr. Godwin Upoki Anywar

Academic Editor

PLOS ONE